# Convert and Speak: Zero-shot Accent Conversion with Minimum Supervision

## ABSTRACT

Low resource of parallel data is the key challenge of accent conversion(AC) problem in which both the pronunciation units and prosody pattern need to be converted. We propose a two-stage generative framework "convert-and-speak" in which the conversion is only operated on the semantic token level and the speech is synthesized conditioned on the converted semantic token with a speech generative model in target accent domain. The decoupling design enables the "speaking" module to use massive amount of target accent speech and relieves the parallel data required for the "conversion" module. Conversion with the bridge of semantic token also relieves the requirement for the data with text transcriptions and unlocks the usage of language pre-training technology to further efficiently reduce the need of parallel accent speech data. To reduce the complexity and latency of "speaking", a single-stage AR generative model is designed to achieve good quality as well as lower computation cost. Experiments on Indian-English to general American-English conversion show that the proposed framework achieves state-of-the-art performance in accent similarity, speech quality, and speaker maintenance with only 15 minutes of weakly parallel data which is not constrained to the same speaker. Extensive experimentation with diverse accent types suggests that this framework possesses a high degree of adaptability, making it readily scalable to accommodate other accents with low-resource data. Audio samples are available at https://convert-and-speak.github.io/demo/.

## CCS CONCEPTS

• Computing methodologies → Accent Conversion.

## KEYWORDS

accent conversion, generative model, speech synthesis

## 1 INTRODUCTION

Accent brings a barrier of understanding when having a conversation between speakers with different accents. The technology of accent conversion aims to break such barriers to make the source accent speakers sound like target accent speakers by changing the pronunciation pattern and prosody while preserving the linguistic content and his/her own speaker identity. This problem is quite challenging since the accent feature affects speech in many aspects such

Permission to make digital or hard copies of all or part of this work for personal or classroom use is granted without fee provided that copies are not made or distributed for profit or commercial advantage and that copies bear this notice and the full citation on the first page. Copyrights for components of this work owned by others than the author(s) must be honored. Abstracting with credit is permitted. To copy otherwise, or republish, to post on servers or to redistribute to lists, requires prior specific permission and/or a fee. Request permissions from permissions@acm.org.
ACM MM, 2024, Melbourne, Australia
© 2024 Copyright held by the owner/author(s). Publication rights licensed to ACM.
ACM ISBN 978-x-xxxx-xxxx-x/YY/MM
https://doi.org/10.1145/nnnnnnn.nnnnnnn

as intonation, rhythm and pronunciation patterns[8]. Take Indian-English for example, they may pronounce $'v'$ as $'w'$ or vice versa, $'th'$ as $'t'$ or $'d'$ and $'p'$ as $'b'$. Besides such pronunciation units difference, the prosody, e.g. intonation and stress is also changed a lot according to the accent. As the example in Appendix(Examples of accent speech), the pitch contour of the Indian-English accent is presented with more ups and downs. Another key challenge is the lack of parallel data. Strictly parallel data with one speaker speaking the same sentence with two different accents barely existed in the public research area.

Some early researches [8] try to explicitly model the pronunciation patterns by building some accent-specific dictionaries to include all possible pronunciations of every word according to the accent type. These methods tend to have poor adaptation because of its assumptions that phonetic knowledge about every accent is available and can be depicted thoroughly in a dictionary and all speakers can be categorized into a few accent clusters are hardly held in the real scenario.

One of the conventional AC methods [5, 13, 28, 30] simplify the accent conversion problem by just converting the voice of the target accent speaker to that of the source accent speaker assuming the utterance with the same content in target accent is available. This kind of method just requires to extract the speaker identity from the source speech without disentangling content with prosody which makes it easier to achieve accent conversion. However, these methods hinder their usage in real applications since the target accent reference is hardly available at conversion stage.

Therefore, reference-free AC methods are more practical and appealing for usage. Some previous approaches [17, 29] try to learn the acoustic mapping between the source accent speech and target accent speech directly with the parallel data in which the same speaker speaks the same content with two different accents. However, such data is extremely rare. So the main idea of this kind of method is to use the voice conversion(VC) technology [14, 24] to synthesize the data set by converting the speaker identity of the target accent speech to that of the source speaker. Such end-to-end mapping-based methods require large amounts of strictly parallel data to achieve a good conversion quality and generalization ability. However, such massive high-quality data can hardly be achieved and the distortions are introduced from the VC stage, even though these VC models have been fine-tuned on the target AC dataset.

To relieve the dependence of parallel data, another kind of approaches [9, 15, 32] which leverage disentanglement technology to remove accent from content, speaker identity, prosody and resynthesize to the target waveform through a synthesizer, e.g. text-to-speech(TTS) model [21]. The synthesizer is trained on the target accent speech to generate speech with prosody in target accent. To remove accent from content and speaker identity, some auxiliary models or tasks are carefully designed, e.g. accent-agnostic automatic speech recognition(ASR) model or phoneme classification

task. Text transcriptions are largely used in these solutions to provide the supervision of accent-agnostic semantic representation. Such two-stage mapping-based methods still require large amounts of (text-accent speech) pairs combined with dedicated auxiliary tasks to achieve accent-agnostic semantic feature and generate diverse speech in target accent.

In this work, we propose a two-stage generative framework with conversion stage and speaking stage to achieve accent conversion. The conversion stage is operated on semantic level by generating the semantic tokens in target accent from source accent. The speaking stage is using a generative-based synthesis model conditioned on the converted semantic tokens to generate the speech with prosody in target accent. Splitting the AC task into these two sub-tasks and realizing conversion with the bridge of semantic tokens which are extracted from raw speech enable the "speaking" module to be independent of parallel data and use massive amount of target accent speech without text transcriptions to generate speech with good quality and diversity in the target accent domain. Meanwhile, it makes it easier for the conversion part to just learn the pronunciation pattern/phoneme difference with a small amount of weakly parallel data which is not constrained to the same speaker.

Both of the stages are seq2seq tasks based on decoder-only Transformer architectures [22]. Inspired by the ideas from machine translation community to reduce the need for supervision, we leverage the BART/T5-style pre-training [12] to significantly reduce the amount of parallel supervision required to train the conversion part. Such pre-training with a pretext task on target-accent data is designed to learn the pattern of generating semantic units, e.g. the joint probability of phonetic units in target accent space. Furthermore, to reduce the complexity and latency of the speech generation process, we design a single-stage autoregressive generative model which generates all vector quantizers(VQs) in one step based on TF-Codec [10].

**Our contributions:**(i)We propose a state-of-the-art generative-based framework for accent conversion which is capable of converting prosody pattern as well as pronunciation units, as evaluated with objective and subjective metrics on public Indian-English accent to general American-English accent conversion test set. (ii)We use the pre-training technology on the conversion part to largely reduce the amount of parallel supervision to only 15 minutes of weakly parallel data. (iii)This framework can be easily extended to other low-resource accents as our experiments on Chinese-English accent and Korean-English accent shown. (iv)We propose a single-stage speech generative model based on TF-Codec with better speech quality and speaker similarity at lower computation cost and latency compared with the multi-stage generation process in other generative models based on Encodec (proposed: 50 AR steps/1 sec of audio vs Encodec-based: 75 AR steps+7 NAR steps/1 sec of audio).

The paper is organized as follows. Section 2 introduces the background of accent conversion and speech generative models. Section 3 introduces the proposed framework in detail. Section 4 validates the performance of the proposed framework mainly on Indian-English accent to general American-English accent and extensive experiments are undertaken to substantiate the efficacy of our model design. Section 5 concludes the paper.

## 2 BACKGROUND

### 2.1 Accent conversion

For accent conversion task, there has not been a public parallel corpus that contains pairs of audios having the same contents yet coming from the same speakers in different accents. So mainly two kinds of methods are proposed to accomplish this task in the literature. One is to synthesize the dataset containing the pairs of audios in the same voice but in two different accents with another voice conversion model and learn the acoustic mapping between them to accomplish accent conversion. [29] build the golden speaker utterance by converting the general American-English speaker's voice to the source-accent speaker's with a pretrained source speaker's synthesizer. Then use this golden speaker utterance as the target to learn the mapping of the mel-spectrogram based on a seq2seq VC system. [17] use a pretrained VC model to build the parallel data and trained the AC model based on Tacotron[21] conditioned on the semantic representation extracted from wav2vec 2.0[1]. This end-to-end mapping-based approach needs large amounts of data to achieve a good zero-shot ability and the auxiliary VC model usually needs to be fine-tuned on the AC data set to alleviate the error caused by voice conversion step. These methods also constrain the output to be generated with the same length of the input which limits the conversion quality since the prosody of the speech is largely affected by the accent.

Another approach is to regard accent conversion as a decomposition and resynthesis task in which the accent is separated from content, speaker identity, prosody and resynthesize to the target waveform in a TTS manner. [15] disentangles different features in multi-stage with several off-the-shelf models. Specifically, an accent-robust ASR model is trained using source accent speech with text labels to separate the source accent from the content. A multi-speaker TTS model with a global speaker encoder is trained with a large corpus of target-accent speech to map the accent-agnostic linguistic features to acoustic features with the voice of source speaker and target accent prosody. Similarly, [32] learns the semantic embeddings directly from the accent speech with the supervision of text embeddings extracted from text labels. Such two-stage mapping-based non-parallel AC approach relieves the burden on the parallel data but needs to leverage large amounts of (text-accent speech) data and dedicated auxiliary tasks. With regard to performance, these methods are not good enough in conversion quality and diversity. It also suffers from poor zero-shot generalization ability with limited high-quality data available. The same accent speaker is used in their training and testing. Another work [9] treats the decomposition and resynthesis in an end-to-end manner. It designs a Pseudo Siamese Disentanglement Network (PSDN) with two streams in which one stream is used to learn the acoustic feature of target accent speech and the other auxiliary stream is used to build the information gap with the target stream to disentangle the content with accent, complemented with another adversarial accent classifier with gradient reversal layer(GRL). This framework can be used in the zero-shot scenario but the performance is not clear since their demo page is out-of-date and the metrics in their paper can not be compared with other public works.

Compared to prior AC approaches, the proposed generative framework neither requires large amounts of parallel data spoken

by the same speaker nor supervision from text labels or auxiliary tasks to achieve a good conversion quality.

## 2.2 Speech generative models

Recently, the speech generative models show large potential in generating contextual consistent, natural and diverse audio/speech based on a speech neural codec with the in-context learning of a referenced prompt. AudioLM[2], designed for zero-shot audio generation, uses Soundstream[26] codes as intermediate representation of acoustic features for speech synthesis. It also shows the strong ability of in-context learning with a short prompt to maintain acoustic information such as speaker identity, prosody style and acoustic environment in the continuations. VALL-E[23], verified in the TTS task, based on Encodec[4] tokens has also shown better zero-shot ability, speech naturalness and diversity than non-generative based TTS models.

For model structure, these models take the decoder-only autoregressive Transformer structure[22] to build the conditional correlation of the acoustic features tokenized by a neural speech codec and the semantic tokens/phoneme sequences. When generating codes, they usually need multiple stages since the neural codecs they use are built on residual vector quantization(RVQ) which consists of a hierarchy of all VQs. In AudioLM, the first several quantizer layers are predicted in the first stage to get the coarse information of the speech and the rest layers are predicted based on the coarse layers to get the fine details of the speech. VALL-E simplifies the generating process by replacing the second stage with nonautoregressive(NAR). In its design, the first quantizer is generated with the AR model and the others are generated with the NAR model(all frames are predicted simultaneously when predicting each codebook) based on the previous quantizers.

Different from these existing generative models, we propose a one-stage AR generative model which generates all VQs in one step to achieve lower complexity and latency as well as better quality.

## 3 PROPOSED FRAMEWORK

### 3.1 Overview

In the proposed framework, as shown in Figure 1, the pronunciation patterns are converted at discrete semantic token level and the prosody is re-synthesized in target accent with a speech generative model. Specifically, we use a pre-trained self-supervised speech representation model, e.g. HuBERT[6] to extract discrete semantic tokens. A neural codec based speech generative model is used to generate the acoustic codes of the codec conditioned on the converted semantic tokens with the style prompt to maintain the source speaker's voice. Both the conversion and generative models are based on autoregressive decoder-only Transformer structure. More details are discussed in Section 3.2 and Section 3.3.

### 3.2 Semantic token conversion

The conversion module is designed as a seq2seq task in discrete semantic token space in which the source accent semantic tokens are converted to the target accent semantic tokens iteratively in an autoregressive manner. To handle the shortage of parallel data, inspired by BART and T5[12, 20], we use large amounts of target accent data to pre-train the conversion module with a pretext task

in our scenario. We then fine-tune the conversion module with a small amount of weakly parallel data.

**Pre-training.** In our scenario, the pretext task is designed to build the probability space of discrete semantic tokens in the target accent domain so that the target accent semantic tokens can be generated according to its context of previous tokens in the target accent domain in a closed-loop manner. In this pretext task, the model is trained in a self-supervised manner which is to produce the original token sequence $Y = \{y_0, ...y_t\}, t < T$ conditioned on the corrupted token sequence $\bar{Y} = \{\bar{y}_0, ...\bar{y}_t\}, t < T$, formulated as

$$p(Y|\bar{Y}; \theta_{AR}) = \prod_{t=0}^{T} p(y_t|y_{<t}, \bar{Y}; \theta_{AR}) \quad (1)$$

We have experimented with corruptions like token masking, token deletion, and token in-filling and we find the token in-filling scheme works the best. Specifically, following the text in-filling scheme in BART[12], a number of text spans are sampled according to $Bernouli(p)$ where $p = 0.5$. The span lengths are drawn from a Poisson distribution($\lambda = 5$). We train the pretext task with large amounts of target accent data which is available in the public corpus.

**Fine-tuning.** Since some phonemes in the source accent need to be converted to the target accent ones, a mapping between these phonemes needs to be learned. Specifically, we fine-tune the pretrained conversion model conditioned on the semantic tokens in source accent with a small amount of weakly parallel accent data. Correspondingly, the training can be formulated as

$$p(Y|X; \theta_{AR}) = \prod_{t=0}^{T} p(y_t|y_{<t}, X; \theta_{AR}) \quad (2)$$

in which $X = \{x_0, ...x_t\}, t < T$ is the source accent semantic token sequence and $Y = \{y_0, ...y_t\}, t < T$ is the target accent semantic token sequence.

### 3.3 Target accent speech generation

The target accent speech generation is achieved by training a separate generative model on a large target accent speech corpus. We design a new speech generative model based on TF-Codec[10]. This model generates acoustic tokens of TF-Codec iteratively through a single-stage causal speech generation, conditioned on the converted/target accent semantic tokens.

*3.3.1 Speech tokenizer with TF-Codec.* We use the pre-trained causal speech neural codec TF-Codec to extract the acoustic token of each frame. Unlike [10], we remove the predictive loop and use the non-predictive model at 6 kbps for efficient acoustic modeling with high-quality output. Specifically, the TF-Codec takes the 16kHz magnitude-compressed time-frequency spectrum with a window length of 20 ms and a hop length of 5 ms as input. Then a stack of 2D causal convolutional layers, followed by a temporal convolutional module (TCM) and a gated recurrent unit (GRU) block is used to capture the short-term and long-term temporal dependencies between the input frames in a causal manner. For the quantization, it combines 4 frames together, producing a frame rate of 50 Hz for quantization. Instead of using RVQ, it employs group quantization where the latent embedding is split into $K$ groups and

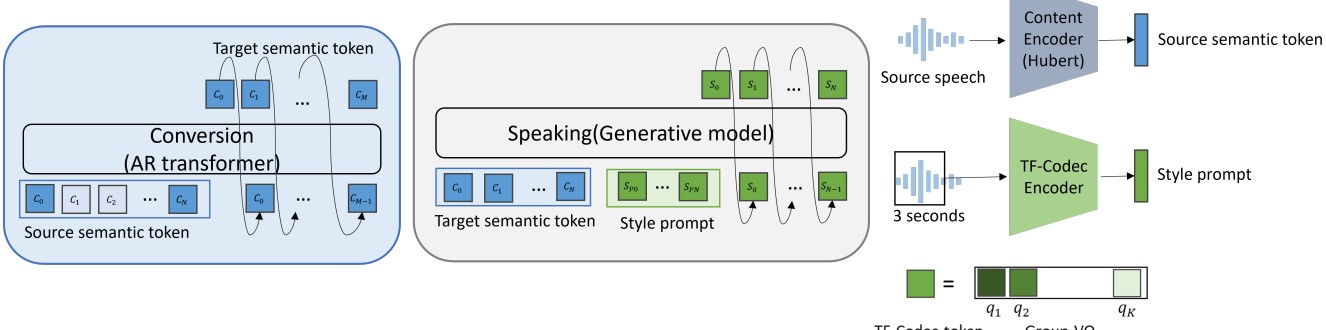

**Figure 1: Proposed framework. the source accent semantic tokens are converted to target accent semantic tokens in the first stage and the speech is generated with target accent prosody conditioned on the converted semantic tokens in the second stage. The style prompt is extracted from the first 3 seconds of the source speech. TF-Codec token is a group of concatenated embeddings of each quantizer.**

each group is quantized by a vector quantizer with a codebook of 1024 codewords. All $K$ acoustic codes are concatenated and decoded to get the reconstructed waveform.

*3.3.2 Single-stage causal speech generation.* As the group quantization in TF-Codec encodes each group independently, we leverage a single-stage causal speech generation to generate acoustic codes of all $K$ quantizers simultaneously for each frame. As shown in Figure 1, TF-Codec token, which is the concatenated embeddings corresponding to all $K$ quantizers, is generated in one-stage autoregressive manner conditioned on the target accent/converted semantic tokens and style acoustic tokens. For each group embedding in TF-Codec token, the dimension is $D_{token}/K$, in which $D_{token}$ is the dimension of the embedding in transformer. $K$ classification heads are employed to predict the $K$ acoustic codes for current frame separately. The training target can be formulated as

$$p(C_:|Y, \widetilde{C}_:; \theta_{AR}) = \prod_{t=0}^{T} p(c_t|c_{<t}, Y, \widetilde{C}_:; \theta_{AR}) \qquad (3)$$

in which $Y = \{y_0, ...y_t\}, t < T$ is the semantic token sequence from target accent speech. $C_:$ is TF-Codec token sequence of target accent speech. $\widetilde{C}_:$ is the TF-Codec token sequence of style acoustic prompt. We do not distinguish $\widetilde{C}_:$ from $C_:$ in training. The concatenation of $\widetilde{C}_:$ and $C_:$ is a whole sequence. During inference, the first 3 seconds of the source speech is used as $\widetilde{C}_:$.

## 4 EXPERIMENTS

To evaluate the performance of the proposed framework, we take Indian-English as source accent and general American-English as target accent which is a common scenario in the research literature.

### 4.1 Experimental Setup

**Dataset.** To train the speech generative model and pre-train the conversion model, LibriTTS dataset [27] is used as our training data. The dataset contains approximately 585 hours of general American-English speech data, sourced from audiobooks available on the public LibriVox project. To fine-tune the conversion model, L1-L2 ARCTIC dataset [11, 31] is used. L1-L2 ARCTIC dataset is a dataset with accent speakers speaking the same content. To build the parallel data, we select a general American-English speaker named "bdl" as the target accent speaker and "ASI" as the Indian-English speaker. Among all their utterances, 1000 utterances, about 50 minutes of speech are used in the training, 50 utterances are used in validation and the remaining 100 utterances are used for testing. To better verify the zero-shot ability, we also add speaker p248 from VCTK dataset and another 4 Indian-Englsh speakers which are not used in training from L1-L2 ARCTIC dataset into testing. To be noted that the 20 utterances from speaker p248 in VCTK are used to compare with the existing machine-learning based AC method [15]. Besides these 20 cases, we add another 20 utterances of each testing speaker in L1-L2 ARCTIC for objective evaluation, e.g. 120 cases in total, and random 8 utterances per speaker for subjective evaluation, e.g. 60 cases in total.

**Model and configuration.** For semantic tokenizer, we employ the HuBERT-Base model[1] and k-means algorithm with 500 clusters to extract semantic tokens. It is trained on LibriSpeech[18] mostly consisting of general American-English. It generates a discrete semantic token sequence at 50Hz framerate for 16kHz audio. Previous studies[7, 19] show that HuBERT is a good representation of speech content and removes most of the speaker identity so that we can use the weakly parallel data in the fine-tuning stage of the conversion module. For the acoustic tokenizer, the number of quantizers in TF-Codec ($K$) is set to 16. The transformer used in conversion model and generative model is the same structure with 12 layers of 16 attention heads, a feed-forward layer with dimension of 4096, and a dropout layer with rate of 0.1. The embedding dimension in transformer($D_{token}$) is 1024. The generative model and pre-training stage of conversion model are trained on 8 NVIDIA TESLA V100 32GB GPUs with a batch size of 4k tokens per GPU. The ScaledAdam[25] optimizer is used. The learning rate is set to

---

[1]https://huggingface.co/facebook/hubert-base-ls960

**Table 1: Evaluation on VCTK test set(20 cases from speaker p248 as Liu. et al's). SPK of accent source is computed on different utterances of the source speaker.**

| Framework | NISQA-TTS(↑) | MOS-Naturalness(↑) | SPK(↑) | MOS-Accent(↑) |
|---|---|---|---|---|
| Accent source | 4.60 | 4.52±0.06 | 0.594 | 0.5% |
| Referenced ground truth | 4.41 | 4.34±0.06 | - | 70.0% |
| Liu. et al [15] | 3.84 | 3.95±0.07 | 0.168 | 67.6% |
| Generative-only model(EnCodec) | 3.65 | 3.84±0.08 | 0.429 | 35.1% |
| Generative-only model(TF-Codec) | 4.10 | 4.00±0.05 | 0.502 | 35.0% |
| Proposed(conversion+generative) | 4.24 | 4.08±0.06 | 0.408 | 69.3% |

**Table 2: Evaluation on L1-L2 ARCTIC test set. LCSR of ground truth speech is calculated between ground truth utterances of different speakers.**

| Framework | NISQA-TTS(↑) | MOS-Naturalness(↑) | SPK(↑) | LCSR(↑) | MOS-Accent(↑) |
|---|---|---|---|---|---|
| Accent source | 3.65 | 4.16±0.07 | 0.641 | 0.545 | 0.5% |
| Referenced ground truth | 3.54 | 4.14±0.08 | - | 0.744 | 79.3% |
| Generative-only model(EnCodec) | 3.32 | 3.79±0.07 | 0.511 | 0.545 | 35.0% |
| Generative-only model(TF-Codec) | 3.59 | 3.91±0.06 | 0.543 | 0.545 | 35.2% |
| Proposed(conversion+generative) | 3.84 | 3.93±0.06 | 0.438 | 0.622 | 74.3% |

0.01, with a warmup for the first 5k steps and decays exponentially. The speech generative model is trained for 500k steps and the conversion model is trained for 100k steps. The fine-tune stage of the conversion model is processed on one GPU of NVIDIA Tesla A100 80GB, with a batch size of 20k tokens. The same optimizer is used. The learning rate is set to $2 \times 10^{-5}$, with a warmup for the first 160 steps. The fine-tuning of the model is trained for 1k steps. During inference, we employ Top-$k$ algorithm to generate each token, in which $k = 2$ for conversion model and $k = 10$ for speech generative model. For each case, we infer 5 times and select the one with best LCSR metric as the choice.

**Baseline models.** To show the superiority of the proposed framework, we select 3 models as our baselines. The existing machine-learning based AC method [15], which is the best model available in the public to our knowledge. The generative-only models without the conversion module are also used as our baselines, in which we compare with the commonly-used EnCodec-based multi-stage generative model and the proposed single-stage TF-Codec based generative model.

**Evaluation methods on accent similarity.** To evaluate the performance of accent conversion, both objective and subjective metrics are used. Intuitively, we use the metric Longest Common Subsequence(LCS) to evaluate the similarity of the converted semantic token sequence and referenced target semantic token sequence. To eliminate the disturbance of the duration of each word, the duplicated tokens in the sequence are removed. To remove the effect of the utterance length in the final average statistics, the Longest Common Subsequence Ratio ($LCSR = LCS/utterance\_length$) is used. The smaller utterance length is used to calculate the LCSR

of the testing pair. We also use the latest state-of-the-art English accent classification model CommonAccent[2] [33] to identify the accent of the synthesized speech. Besides, we conduct a subjective A/B testing in which participants are asked to choose the one that sounds more close to general American-English accent in an A/B pair. Each A/B pair contains cases chosen from any two of the competitors. The accent source and ground truth are also included to ensure the validity of the testing. To remove the potential factor of speaker identity in the subjective testing, in Table 2, both the accent source and referenced ground truth are chosen from multiple speaker's utterances, e.g. 5 Indian-English speakers and 4 general American-English speakers from L1-L2 ARCTIC test set. To be noted, the referenced ground truth is used here with the same sentence spoken by different speakers. Particularly, the participants are trained to distinguish the accent difference by listening to several pairs of <Indian-English, general American-English> samples before the formal testing. 20 participants who are proficient in general American-English are invited to conduct these evaluations. MOS-Accent, the percentage of being selected as general American-English accent is used as the metric of this subjective testing.

**Evaluation methods on speaker similarity and speech quality.** To evaluate the speaker identity maintenance, the speaker similarity metric is calculated as the cosine similarity of the two speaker vectors, which are extracted from the source accent speech and the

---

[2]https://huggingface.co/Jzuluaga/accent-id-commonaccent_ecapa

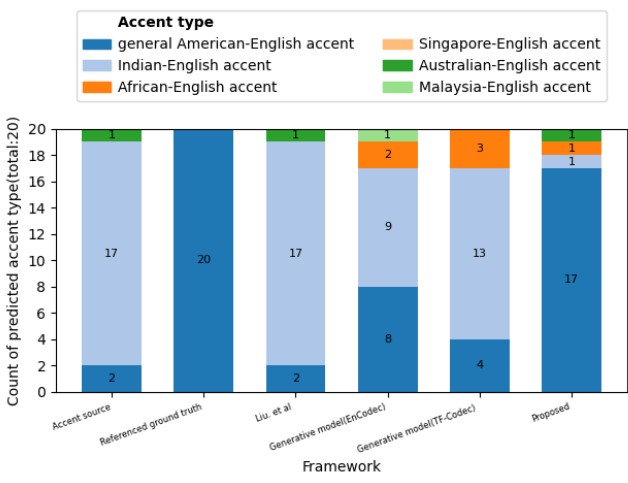

**Figure 2: Accent classification results for VCTK test set, evaluated by CommonAccent.**

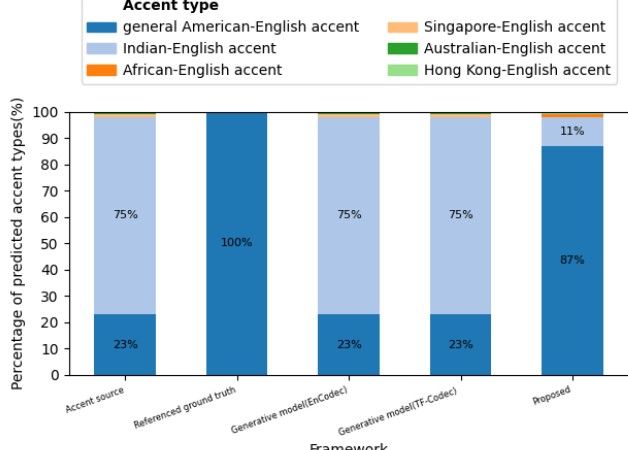

**Figure 3: Accent classification results for L1-L2 ARCTIC test set, evaluated by CommonAccent.**

converted speech, correspondingly. WavLM-TDNN[3] [3], a state-of-the-art speaker verification model, is used to get the speaker vector from a speech. To evaluate the naturalness, we use NISQA-TTS[16], which is commonly used for synthesized speech. We also conduct MOS testing, in which the raters are asked to give a score ranging from 1 (lowest quality) to 5 (highest quality) according to the overall subjective quality. The MOS-Naturalness with confidence level of 95% is used as the metric.

### 4.2 Results

**Accent similarity.** As shown in Table 1 and Table 2, the MOS-Accent metric of the proposed framework on both datasets ranks the highest and very close to the ground truth. Compared with the generative-only models, e.g. Generative-only model(EnCodec) and Generative-only model(TF-Codec), the proposed framework highly surpasses them for accent conversion performance, indicating the effectiveness of the conversion module. This is also verified by the LCSR metric on L1-L2 ARCTIC test set in Table 2, where the proposed framework closely approaches the ground truth LCSR after conversion. The analysis on HuBERT with accent input in Section 4.5.1 also shows the HuBERT tokens are affected by the source Indian-English accent, especially for those accent speech with phonetic changes, indicating the necessity of the semantic conversion module. Figure 2 and Figure 3 show the accent classification results from CommonAccent. Compared with other methods, the proposed framework converts most of the Indian-English accent input to the target general-American English accent. In Figure 3, for the rest of 11% cases which are identified as Indian-English accent, besides classification error, most of the cases are short and the conversion quality of some cases is not good enough which leaves room for further improvement on the robustness of the generative framework. For Liu. et al [15]'s method, it is interesting to find that there is a big gap between classification metric(as bad as accent

source) as shown in Figure 2 and MOS-Accent(relatively good) as shown in Table 1. We think the bad result in terms of classification metric comes from its poor conversion ability on the pronunciation units. Most of the pronunciation units are not converted well, e.g. the pair of $('b', 'p')$. For prosody conversion, the quality is relatively good compared with the source accent, which contributes to the high subjective metric. The audio samples can be found in our demo page. Overall, the proposed framework achieves much better accent conversion performance as verified by both objective classification metric and subjective metric.

**Speech quality and Speaker similarity.** According to the NISQA-TTS and MOS-Naturalness metric in Table 1 and Table 2, the proposed framework ranks at the top level. Compared with Liu's method, the proposed framework achieves much better speech quality and speaker similarity. Artifacts of Liu's model can be found in some cases, as shown in the demo page. What's more, we find the better speech quality and speaker similarity can be achieved with TF-Codec based generative model according to the comparison of Generative-only model(TF-Codec) and Generative-only model(EnCodec). This can also be verified by our demo cases. Compared with Generative-only model(TF-Codec), the SPK value of the proposed framework drops a bit but the subjective judgement on the demo cases are quite similar. We think this is caused by the error from the speaker vector extractor WavLM-TDNN. Since WavLM-TDNN is not trained on accent speech so the extracted vector contains not only the speaker identity but also accent information. So with better accent conversion, the speaker vector of the converted speech and the accent source speech tend to be more different, resulting in a lower cosine similarity. It should be more reasonable to use this metric to compare Generative-only model(EnCodec) with Generative-only model(TF-Codec) and the proposed with Liu's method since both of them are with/without accent leak in the converted speech.

---

[3]https://github.com/microsoft/UniSpeech/tree/main/downstreams/speaker_verification

**Table 3: Complexity comparison of speech generative module.**

| Framework | Model parameters(M) | Decoding steps(/s) |
|---|---|---|
| Generative-only model(EnCodec) | 262.3 | 75 AR + 7 NAR |
| Generative-only model(TF-Codec) | 100.8 | 50 AR |

**Table 4: Accent conversion quality with parallel data of 50 mins, 30 mins, 15 mins.**

| Parallel data amount | MOS-Accent(↑) |
|---|---|
| 50 mins | 59.3% |
| 30 mins | 58.1% |
| 15 mins | 56.8% |

## 4.3 Efficiency of single-stage causal speech generation

Here we compare the complexity of the proposed single-stage causal speech generation scheme based on TF-Codec with the multi-stage speech generation scheme based on Encodec. In the Encodec-based generative models, two stages are usually taken, with a combination of autoregressive(AR) stage to generate the first quantizer and NAR stage to generate the rest of the quantizers of all time steps based on the previous quantizers. The Encodec used in the experiment is composed of 8 quantizers with the frame rate of 75 Hz and sample rate of 24kHz. The complexity is shown in Table 3 in terms of model parameters and decoding steps. According to Table 3, TF-Codec based generative model saves more than 50% in model size and takes a pure causal decoding scheme with fewer steps.

## 4.4 Training with minimum supervision

In this section, we further reduce the parallel data used in the fine-tuning stage of the conversion model, from 50 minutes (proposed in Table 1 and Table 2) to 30 minutes and 15 minutes, respectively. We use the MOS-Accent as the evaluation metric and test on the VCTK test set. We also add the baseline models into A/B testing for comparison. As shown in Table 4, by decreasing the data amount, the performance drop is negligible. With the minimum supervision of 15 minutes, the performance is still relatively good, which shows its high potential for extension on other accents with low-resource data, such as Chinese-English accent and Korean-English accent to general American-English accent cases in Section 4.7.

## 4.5 Supportive analysis

*4.5.1 HuBERT tokens from accent speech.* In this section, we evaluate how accent affects the HuBERT tokens. Specifically, we build a parallel data set from L1-L2 ARCTIC dataset in which the Indian-English speaker and general American-English speaker speak the same content. Both of them are fed into the HuBERT model used in the paper to get the semantic token sequence. The LCSR metric is used to evaluate the content similarity between these two HuBERT token sequences. 1000 pairs are used in this experiment. To further study the phoneme change effect, the accent cases are divided into the one with phoneme changes and without phoneme changes. Since the speakers of each pair are different, the effect of the speaker identity on the LCSR metric is also calculated as a reference. As Table 5 shows, with the source accent introduced, HuBERT tokens have changed a lot, degrading from 0.747 to 0.569 in terms of LCSR. For those cases with specific phoneme changes, more tokens have been changed from the target accent references(LCSR: 0.541).

*4.5.2 Accent effect on the style prompt in speaking module.* We use the accent source as the style prompt in the speech generative model to extract speaker identity of the source speaker. This section is to evaluate if the accent feature will be extended through the in-context learning in the speech generative model. We conduct empirical study to substantiate such usage. Specifically, we design an A/B testing to compare the accent similarity of the synthesized speech generated with two kinds of prompts in different accents conditioned on the same content. For testing, we take Indian-English and general American-English as two prompt types for comparison. We build 100 pairs of samples to test. Each pair contains an utterance in general American-English accent which is used to extract HuBERT semantic tokens, an utterance from a general American-English speaker and from an Indian-English speaker working as the style prompts. The prompts are cut to 3 seconds. Examples can be found in our demo page. For subjective testing, 20 participants who are college students majoring in American-English are asked to distinguish the two synthesized speech and choose the one which sounds closer to general American-English. The percentage of being selected as general American-English accent is used as the evaluation metric. We also use the CommonAccent to identify the two synthesized speeches. As Table 6 shows, no matter general American-English or Indian-English prompt type, the percentage to be selected as general American-English by users is about 50% and almost all the synthesized speeches are identified as general American-English. The similar results are observed from the testing with more accent prompt types as Chinese-English accent and Korean-English accent in Appendix(In-context learning with more accent prompt types). Furthermore, we find accent prompts do have effect on the prosody modeling but are quite limited. According to our experiments on the effect of accent prompt length in Appendix(Effects of accent prompt length), with the prompt in Indian-English accent becomes longer, increased from 3s to 7s, the percentage of predicted general American-English accent drops from 84% to 73%. So we can use 3 seconds of accent source as a prompt to catch the source speaker's identity without bringing the source accent back to the converted speech.

## 4.6 Ablation Study

**Decoupling design** We compare with the solution in which the parallel data is used to fine-tune the generative-only model directly. In such a way, the model is guided to learn the phoneme and prosody conversion simultaneously and blindly through an AR Transformer model. As Table 7 shown, the LCSR of generative-only

**Table 5: HuBERT tokens from accent speech. Lower LCSR means less similarity between source accent speech and target accent speech. Source accent: Indian-English. Target accent: general American-English.**

| Influencing factors | LCSR($\uparrow$) |
|---|---|
| Speaker identity | 0.747 |
| Accent without phoneme changes(w. speaker identity change) | 0.569 |
| Accent with phoneme changes(w. speaker identity change) | 0.541 |

**Table 6: Comparison of the synthesized speech with two accent prompt types: general American-English and Indian-English. CommonAccent metric shows the percentage of being predicted as general American-English. A/B testing metric shows the percentage of being selected as general American-English in the A/B pair.**

| Prompt type | CommonAccent | A/B testing |
|---|---|---|
| General American-English accent | 98% | 50% |
| Indian-English accent | 97% | 50% |

**Table 7: Ablation study on the decoupling design.**

| Framework | LCSR($\uparrow$) |
|---|---|
| Proposed(w. decoupling) | 0.622 |
| w.o decoupling | 0.020 |

**Table 8: Comparison of the semantic conversion quality w./w.o pre-training.**

| Framework | LCSR($\uparrow$) |
|---|---|
| Proposed (w. pre-training) | 0.622 |
| w.o pre-training | 0.103 |

**Table 9: Results on general American-English accent conversion from Chinese-English accent and Korean-English accent. CommonAccent metric shows the percentage of being predicted as general American-English.**

| Source accent type | CommonAccent |
|---|---|
| Chinese-English accent | 95% |
| Korean-English accent | 100% |

model without decoupling on L1-L2 ARCTIC test set is quite low at 0.02, indicating most of the content has been destroyed and the model fails to learn such mapping with so little amount of parallel data.

**The effect of pre-training for semantic conversion module** To verify the validity of the language pre-training technology used for the semantic conversion module, we compare it with the solution where the conversion model is trained from scratch with the weakly parallel data. All parallel data(about 50 mins) are used for training. As Table 8 shows, without pre-training, the results degrade by a large margin. This is reasonable since the pre-training stage lays a good foundation for the fine-tuning stage to just focus on learning few semantic units which are different in the two accents.

## 4.7 Extensions to more source accent types

We extend the proposed framework to Chinese-English accent and Korean-English accent. Specifically, another 15 minutes of weakly parallel data of <Chinese-English accent, general American-English accent> and <Korean-English accent, general American-English accent> from L1-L2 ARCTIC dataset is used in the fine-tuning stage of the conversion module. The conversion accuracy of Chinese-English accent is 95% and Korean-English accent is 100%, identified by the CommonAccent metric as shown in Table 9. Audio samples can be found in our demo page.

## 5 CONCLUSIONS

In this work, we propose a two-stage generative framework for accent conversion task in which the conversion is operated on the semantic token level and synthesized to a target accent speech with TF-Codec based generative model. Experimental results show the proposed framework achieves the state-of-the-art performance in terms of accent similarity, speech quality and speaker maintenance with limited parallel data. With the language pre-training technology, only 15 minutes of parallel data, not constrained to the same speaker reaches to a good conversion quality, which shows large potential for an easy extension for other accents with low-resource data. The proposed single-stage AR generative model achieves better speech quality at lower complexity, which can be used for other speech generative tasks. In the future, we will further improve the robustness of the generative framework for AC task.

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
