# OpenReview forum: "Convert and Speak: Zero-shot Accent Conversion with Minimum Supervision"
_acmmm.org/ACMMM/2024/Conference — MM2024 Poster_

### Official Review · Reviewer_Lxfq · 2024-05-20

**Rating:** 5
**Confidence:** 4

**Summary:**

This paper addresses the challenge of accent conversion (AC) with limited parallel data. The proposed "convert-and-speak" framework tackles this by decoupling the conversion and speech generation processes. Conversion is performed only at the semantic token level, enabling the use of large amounts of target accent speech for speech synthesis. This design eliminates the need for parallel data for the "conversion" module and reduces the requirement for transcribed data. Furthermore, leveraging language pre-training technology further minimizes the need for parallel accent speech data.

To address complexity and latency, a single-stage AR generative model is employed for speech generation, achieving both high quality and efficient computation.

**Strengths:**

The paper introduces a novel two-stage framework that effectively tackles the data scarcity issue in accent conversion. This approach leverages the power of pre-trained language models and large target-accent speech datasets, making it highly efficient and adaptable.
Semantic Token Level Conversion: It reduces the dependence on parallel data and allows for leveraging large-scale language models for more robust and accurate conversion.
Decoupled Design:  It enables the use of separate and optimized models for each task, leading to improved performance and scalability.
Single-Stage AR Generative Model: The use of a single-stage AR generative model for speech synthesis reduces complexity and latency without compromising quality, making the system more practical and efficient.

**Limitations:**

Semantic Token Representation: The conversion process at the semantic token level might not fully capture nuanced prosodic features that are crucial for natural and accurate accent conversion. This could lead to a loss of fine-grained linguistic information during the conversion process.
So,
1. Lack of Analysis on Acoustic Fine-grained Loss from HuBERT:
The paper proposes a novel "convert-and-speak" framework for accent conversion, incorporating the HuBERT model for speech representation. However, the analysis lacks a deeper investigation into the specific contributions of HuBERT's acoustic fine-grained loss to the overall performance. It would be beneficial to delve into how HuBERT's fine-grained acoustic features, particularly its ability to capture phonetic details, contribute to the accuracy and naturalness of the accent conversion.

2. Absence of Coarse-level Evaluation Metrics:
While the paper evaluates accent similarity and speech quality, it lacks an assessment of the coarse-level performance of the conversion system. This is crucial for understanding how well the framework translates the overall meaning and intent of the utterance. Metrics like Word Error Rate (WER) could be used to gauge the accuracy of the conversion system in capturing the semantic content of the speech.

**Suitability:**

3

---

### Official Review · Reviewer_inc7 · 2024-05-21

**Rating:** 3
**Confidence:** 3

**Summary:**

The paper proposes a two-stage generative framework for accent conversion named “Convert and Speak.” The primary objective is to tackle the challenge of converting both pronunciation units and prosody patterns in speech from one accent to another with minimal supervision and data requirements. The approach is structured as follows:
1.	Semantic Token Conversion: The first stage involves converting the semantic tokens of the source accent to the target accent using a pre-trained self-supervised speech representation model, such as HuBERT.
2.	Speech Generation: The second stage involves synthesizing speech in the target accent using a single-stage autoregressive generative model based on TF-Codec.
The framework aims to reduce dependency on large amounts of parallel data and text transcriptions by leveraging pre-training techniques. Experiments demonstrate state-of-the-art performance in converting Indian-English to American-English, with the framework showing adaptability to other accents like Chinese-English and Korean-English.

**Strengths:**

1. Minimal Data Requirement: The proposed method significantly reduces the need for parallel data, achieving high-quality accent conversion with only 15 minutes of weakly parallel data.
2. Generative Framework: The use of a two-stage generative framework efficiently decouples the tasks of pronunciation and prosody conversion, improving the flexibility and scalability of the model.
3. Pre-training Benefits: Incorporating pre-training (e.g., BART/T5-style) helps in reducing the amount of supervised data required for fine-tuning, thus enhancing the model’s performance with limited resources.
4. State-of-the-Art Performance: The framework achieves superior results in accent similarity, speech quality, and speaker maintenance compared to existing methods.

**Limitations:**

1. This work applies the commonly used strategies from text-to-speech. The Minimum Supervision idea follows Google's SPEAR-TTS [1]. The overall framework is very similar to SPEAR-TTS.
2. This paper propose a single-stage generation (parallel prediction all of tokens in one frame), which can improve the inference speed. But such strategy has been demonstrated that it may result in suboptimal generation performance in MusicGen [2] and UniAudio [3].
3.  Dependence on Specific Models: The performance heavily relies on the quality of the pre-trained models like HuBERT and TF-Codec, which might not be equally effective for all types of accents or languages.
4. Evaluation Scope: The evaluation is primarily focused on Indian-English to American-English conversion. Although some other accents were tested, a more extensive evaluation across diverse accents and languages would strengthen the findings.

In summary, this paper gives us a empirical study. From the machine learning perspective, this paper does not give enough insight or contributions. From the Accent Conversion task perspective, this paper gives a good results for this direction. If the authors willing to open-source their checkpoints to the community, it will be a good contribution.

 I am happy to further discuss this paper with authors and other reviewers in the rebuttal stage.

[1] Kharitonov E, Vincent D, Borsos Z, et al. Speak, read and prompt: High-fidelity text-to-speech with minimal supervision[J]. Transactions of the Association for Computational Linguistics, 2023, 11: 1703-1718.

[2] Copet J, Kreuk F, Gat I, et al. Simple and controllable music generation[J]. Advances in Neural Information Processing Systems, 2024, 36.

[3] Yang D, Tian J, Tan X, et al. Uniaudio: An audio foundation model toward universal audio generation[J]. arXiv preprint arXiv:2310.00704, 2023.

**Suitability:**

2

---

### Official Review · Reviewer_fqeh · 2024-06-01

**Rating:** 6
**Confidence:** 3

**Summary:**

The paper introduces "Convert-and-Speak," a two-stage generative framework for accent conversion using minimal supervision. The framework decouples the process into semantic token conversion and speech synthesis in the target accent, significantly reducing the need for parallel data. Experiments show state-of-the-art performance in accent similarity, speech quality, and speaker maintenance using only 15 minutes of weakly parallel data.

**Strengths:**

Reduces dependency on parallel data.
High performance in accent conversion with minimal supervision.
Extensible to other low-resource accents.

**Limitations:**

Some quality loss with reduced training data.
Limited robustness in handling diverse accent challenges.
Needs further improvement for real-world applicability.

**Suitability:**

3

---

### Official Review · Reviewer_JmPy · 2024-06-03

**Rating:** 4
**Confidence:** 4

**Summary:**

This paper proposes a novel two-stage framework for accent conversion, aiming to transform speech from a source accent to a target accent. The first stage converts semantic tokens extracted from the source accent speech into target accent tokens. The second stage employs a generative model to synthesize speech in the target accent, conditioned on the converted semantic tokens.

**Strengths:**

1) The framework leverages pre-training techniques to significantly reduce the amount of parallel data required for training.
2) The proposed single-stage autoregressive generative model for speech synthesis offers improved efficiency compared to existing multi-stage models.

**Limitations:**

The paper presents a promising approach to accent conversion, but it also has some limitations:
1) Although the authors aim to preserve speaker identity, the evaluation of speaker similarity relies on a speaker verification model not trained on accented speech. This could lead to inaccuracies in assessing speaker identity maintenance, as the model might be sensitive to accent-related features.
2) While subjective A/B testing is conducted, the participant pool is limited to college students majoring in American English. This might introduce bias in the evaluation, as their perception of accent similarity might differ from that of a more diverse group.

**Suitability:**

3

---

### Official Review · Reviewer_cxG1 · 2024-06-07

**Rating:** 4
**Confidence:** 3

**Summary:**

This paper proposes a two-stage framework for accent conversion, by generating semantic tokens in the conversion stage and generating speech conditioned on the semantic tokens (extracted by HuBERT) in the speaking stage, getting rid of requirements of large amounts of parallel data and text transcriptions.

The proposed framework is demonstrated effective in converting prosody and pronunciation patterns. Pre-training technologies are adopted to reduce requirement of parallel data by training a Transformer-based decoder for masked token prediction. The decoder is further trained to convert source accent tokens to target accent tokens. The pretrained casual speech neural codec TF-Codec is used to extract acoustic tokens, which are further converted to speech with a single-stage causal speech generation.

**Strengths:**

This paper investigate the novel two-stage framework that leverages pretraining and finetuning to address the data scarcity problem in accent conversion. Experiments on two corpora demonstrate the effectiveness of the proposed framework.

**Limitations:**

The term “stage” is used to refer the “convert” and “speak” stages, and also used to refer to single stage casual speech generation. It would be clearer to use two different terms for them.

In Table 1 and Table 2, MOS-Accent for the reference ground truth is only 70% and 79.3%. Why is it so low for the two datasets? Because the other accent sources are easy to be discerned (hence, MOS-Accent for accent source is as low as 0.5%). Any analysis on this?

The current evaluations can support the effectiveness of the proposed framework in accent conversion. It would be helpful to add more analysis on the prosody patterns and pronunciation units conversion, e.g., by visualization of pitch contours, phonemes, to show the proposed framework’s advantages on prosody and pronunciation conversion.

**Suitability:**

2

---

### Official Review · Reviewer_rZhF · 2024-06-10

**Rating:** 4
**Confidence:** 3

**Summary:**

The paper introduces a two-stage generative framework for zero-shot accent conversion, focusing on efficient use of limited parallel data. The approach decouples the conversion of semantic tokens from speech generation, which allows for leveraging massive amounts of non-parallel data to train a target accent speech synthesis model. This framework achieves state-of-the-art results in accent conversion tests between Indian-English and General American-English.

**Strengths:**

The method introduces innovative techniques for handling semantic tokens in accent conversion, reducing the reliance on extensive parallel datasets, which is a significant advance over previous approaches. The proposed method is particularly valuable for real-world applications where parallel data are scarce but there is a need for high-quality accent conversion.

**Limitations:**

For accent conversion applications, particularly those intended for real-time use, the efficiency of the model during inference is crucial. The paper does not discuss the Real-Time Factor (RTF) or the inference time required by the proposed system.

**Suitability:**

3

---

### Meta-Review · Area_Chair_f3N6 · 2024-07-10

**Recommendation:** Accept (Poster)
**Confidence:** 4

**Metareview:**

The authors propose a two-stage generative framework "convert-and-speak" in which the conversion is only operated on the semantic token level and the speech is synthesized conditioned on the converted semantic token with a speech generative model in target accent domain. The decoupling design enables the "speaking" module to use massive amount of target accent speech and relieves the parallel data required for the "conversion" module. Conversion with the bridge of semantic token also relieves the requirement for the data with text transcriptions and unlocks the usage of language pre-training technology to further efficiently reduce the need of parallel accent speech data.

Strengths:
- The paper significantly alleviates the need for parallel data for accent conversion and is highly relevant for real world applications.
- The use of a two-stage generative framework efficiently decouples the tasks of pronunciation and prosody conversion, improving the flexibility and scalability of the model.
- The framework achieves strong results in accent similarity, speech quality, and speaker maintenance compared to existing methods.

Weaknesses:
- Most of the issues pointed by reviewers have been addressed by the rebuttal.